# Parental Psychological Control and Children’s Prosocial Behavior: The Mediating Role of Social Anxiety and the Moderating Role of Socioeconomic Status

**DOI:** 10.3390/ijerph191811691

**Published:** 2022-09-16

**Authors:** Weida Zhang, Guoliang Yu, Wangqian Fu, Runqing Li

**Affiliations:** 1School of Education, Renmin University of China, 59 Zhongguancun Ave., Haidian District, Beijing 100872, China; 2Faculty of Education, Beijing Normal University, 19 Xinjiekouwai Ave., Haidian District, Beijing 100875, China; 3School of Philosophy and Social Development, Shandong University, Jinan 250012, China

**Keywords:** parental psychological control, prosocial behavior, social anxiety, SES

## Abstract

Complementing internalizing and externalizing developmental outcomes of parental psychological control, in this study, we shift the focus to children’s prosocial behaviors. Drawing on self-determination theory and problem-behavior theory, this study addresses the relationship between parental psychological control, social anxiety, socioeconomic status (SES), and children’s prosocial behavior. The parental psychological control scale, social anxiety scale for children, and prosocial behavior were applied in the study. Participants were 1202 elementary school-age children in China. The present study showed that parental psychological control was negatively associated with prosocial behavior and social anxiety played a partial mediating role between parental psychological control and prosocial behavior. Meanwhile, SES moderated the relationship between parental psychological control and prosocial behavior. The effect of parental psychological control on prosocial behavior was more significant among students with low levels of SES than the higher ones. The findings showed that parenting plays an essential role in the development of children’s prosociality.

## 1. Introduction

Prosocial behavior refers to all behaviors that conform to social expectations and benefit others and society. It plays a vital role in ensuring the survival, development, and well-being of individuals and promoting social stability, harmony, and prosperity [1,2]. For individuals, the development of prosocial behaviors is an important basis for the formation of personality and the establishment of interpersonal relationships, which is important for their mental health, happiness, and social development in adulthood. An empirical study found that people assigned to perform five acts of kindness a day for six weeks reported greater happiness than those in a control group who did not act [3], and spending money on others makes you happier than spending money on yourself [4]. Prosocial behavior can promote individuals to have meaning in life and participating in voluntary activities can enhance an individual’s sense of purpose in life [5]. From the social level, individual prosocial behavior represents positive social value; it is the embodiment of social commonsense and a sense of responsibility, and it is an important basis for building a harmonious society [6]. Childhood is an important stage for the development of prosocial attitudes and behaviors [7]. Therefore, it is of great practical significance to pay attention to children’s prosocial behavior and its influencing factors for cultivating and shaping individual prosocial behavior.

According to Bronfenbrenner’s ecosystem theory, the family is an important micro-environment for a child life, and the parenting style carried out by parents is of great significance to the personality and social development of children [8,9]. Children’s prosocial behaviors come from different motives, including autonomy motivation, altruism motivation, etc., which are affected by parenting style to different degrees [10]. As one of the best-known typologies of parenting styles, four parenting styles have been identified, including authoritative, authoritarian, permissive/liberal, and indifferent (also named uninvolved and neglectful style) [11,12], which are based on the levels of parental affection and control [13]. Parental psychological control means that parents exert influence on their children by controlling or manipulating emotions or thoughts [14].

Self-determination theory posits that certain strategies of motivating behavior in parenting are more or less effective in eliciting behavioral change [15]. Meanwhile, self-determination theory specifies that perceived control by parents would be associated with more need frustration [16]. Under the background of Chinese collectivist culture, parents usually show a high level of psychological control in the process of raising children under the guidance of the parenting concept of “deep love and deep responsibility”. Huang [17] analyzed the data of the “Chinese Education Tracking Study” (CEPS) 2014–2015 and found that the current education mode of Chinese parents is dominated by authoritarian and neglect-oriented styles. Barber found in his research that parental psychological control directly destroys children’s autonomy development by intruding on their children’s inner world and asking them to accept and internalize their own values [18], and then causes a series of internalization problems such as anxiety, depression, and aggressive behavior [19]. Pinquart’s findings suggest that parenting in a controlling (i.e., pressuring, intrusive, and dominant) rather than an autonomic supportive (i.e., allowing choice, initiative, and thinking from the child’s perspective) manner may have a damaging effect on the child’s resilience [20]. In addition, parental psychological control may have adverse effects on children’s prosocial behaviors [21].

Drawing on affective event theory (AET), positive and negative occurrences in daily life can elicit affective reactions [22]. Previous studies have shown that parental psychological control can have an impact on children’s social anxiety. Rork and Morris (2009) [23] found moderate associations between parental control and child anxiety symptoms. Nelemans et al. conducted a longitudinal follow-up study for four years and found that children’s social anxiety symptoms were significantly positively correlated with mothers’ self-reported higher levels of psychological control and lower levels of autonomous support [24]. Therefore, the level of parental psychological control is an important predictor of children’s social anxiety symptoms.

Social anxiety is a kind of negative emotion which is usually negatively correlated with individual prosocial behavior [25]. In daily life, individuals with social anxiety are more likely to be alone than to interact with others, which greatly reduces their opportunities to engage in prosocial behaviors. Vroling et al. found that with increasing levels of social anxiety, the reward felt by rejected individuals would decrease and would lead to a stronger avoidance tendency [26]. Thus, individuals with social anxiety may exhibit decreased levels of prosocial behaviors following explicit exclusion and avoidance tendencies [27,28].

Socioeconomic status (SES) refers to the social status of an individual or group in terms of income, education, and occupational prestige [29]. It not only reflects the social material resources owned by family members and the perception of their social status but also has a significant negative impact on individual psychology and behavior, such as the preference for beauty [30], subjective well-being [31], health [32], and cognitive performance [33]. There is preliminary evidence suggesting the important but complex role that SES plays in associations between parental psychological controls and prosocial behavior in children and adolescents. The family stress model points out that family economic stress (low family SES) can increase the psychological stress of family members [34]. Parents from higher SES backgrounds are able to invest more in the resources and experiences that promote children’s development [35,36]. In families with higher SES, parents tend to adopt a positive parenting style in the process of interacting with their children, while in families with lower SES, parents tend to use a negative parenting style characterized by authoritarian and neglectful parenting [37]. In addition, a longitudinal study revealed that low family SES was associated with low socioemotional resilience [38], which may protect children from stressful environments (e.g., parental psychological control).

### The Current Study

Although a large number of studies have explored the influence of parental psychological control on children’s behavioral outcomes, the outcome variables involved in these studies are mainly negative aspects, such as anxiety and depression, while positive aspects, such as children’s prosocial behaviors, have received relatively little attention. There are some limitations in the previous studies on the relationship between parental psychological control and children’s prosocial behavior. Firstly, most studies were conducted in Western countries, where the parental style is closely associated with cultural context. Therefore, it is worthwhile to examine it in Chinese culture. Secondly, the association between parental psychological control, social anxiety, and children’s prosocial behavior was explored separately, which makes it hard to understand the inner mechanism among them. Thirdly, the influence of SES was largely neglected in the existing studies. To address these gaps in conducted research, we examined the relationship between parental psychological control and children’s prosocial behavior and the effect of social anxiety and SES. Based on self-determination theory and literature evidence, we formulated a hypothetical relationship model (Figure 1). The research hypotheses were as follows (Figure 1):

**H1.** 
*Parental psychological control is negatively correlated with children’s prosocial behavior;*


**H2.** 
*Parental psychological control is negatively correlated with children’s social anxiety;*


**H3.** 
*Social anxiety plays a mediating role between parental psychological control and prosocial behavior;*


**H4.** 
*SES moderates the relationship between parental psychological control and children’s prosocial behavior.*


## 2. Method

### 2.1. Participants

The participants were 1202 students from four ordinary primary schools in Beijing. Among them, 589 (49.0%) were boys, and 613 (51.0%) were girls. The average age was 9.04 (±0.23) years. There were 194 students in grade 4, with an average age of 9.74 (±0.68) years. There were 481 fifth-grade students, with an average age of 10.49 (±0.55) years. There were 302 students in the sixth grade, with an average age of 11.79 (±0.41) years.

### 2.2. Measurements

**Parental Psychological Control Scale.** This scale was compiled by Wang et al. (2007) [39] and was revised from previous studies [40]. Previous studies proved that this scale has good reliability and validity among Chinese students [41]. The 20-item scale measures parental manipulation of their children’s thoughts, behaviors, and emotions (e.g., “When I talk to my mother, my mother always controls the content of the conversation and makes me follow her opinion”). Students rated each question on four points (1 = strongly disagree, 4 = strongly agree). The higher the score, the higher the level of parental psychological control. The parental psychological control scale demonstrated high Cronbach’s alphas across the different samples, ranging from 0.85 to 0.91 [42]. In this study, the coefficient of internal consistency of the two sub-dimensions was 0.94 and 0.91, respectively.

**Social Anxiety Scale for Children**. The Social Anxiety Scale for Children (SASC) [43], a 10-item self-report measure, was used to assess levels of anxiety experienced before and during situations of social interaction (e.g., “I worry about doing something new in front of other kids.”) that participants were asked to rate on a three-point scale ranging from never true (0) to sometimes true (1) to always true (2). The SASC demonstrated high internal consistency from previous literature, Cronbach’s alphas across the different studies were high (i.e., 0.90–0.92) [44], and in the present study, they were excellent (α = 0.93).

**Prosocial Behavior Scale.** The Prosocial Behavior Scale was selected from the Strengths and Difficulties Questionnaire (SDQ). The SDQ was designed and developed by American psychologist Goodman in 1997 according to the Diagnostic and Statistical Manual of Psychiatry, Fourth Edition (DSM-IV), and the Mental and Behavioral Classification, Tenth Edition (ICD-10) diagnostic criteria. The SDQ includes five factors, including emotional symptoms, conduct problems, hyperactivity, peer interaction problems, and prosocial behavior, with a total of 25 items. Prosocial behavior subscales contain five topics (e.g., “I often share things with others, such as food, toys, and pen”), and each subject adopts three-point scoring (0 = is not correct, 1 = some right, 2 = all correct). The SDQ showed good internal reliability, with alphas ranging from 0.66 to 0.82 for all subscales [45]. In this study, the coefficient of internal consistency was 0.82.

**Socioeconomic Status (SES) Scale.** The following five indicators were used to measure SES in this study: father’s occupation, mother’s occupation, father’s education level, mother’s education level, and family income. The collected indicators were processed in the following steps: (1) The parent with a higher educational level and occupational status was selected to be included in the calculation and merged into three indicators: annual family income, the highest educational level of the parents, and the highest occupational social status of the parents. (2) To deal with the missing values in the three indicators, and (3) by referring to previous studies, the original scores of indicators were converted into standard scores, and principal component analysis was conducted to obtain the comprehensive SES index: SES = (0.65 × Z annual family income + 0.78 × Z parents’ highest education level + 0.86 × Z parents’ highest occupational status)/1.77. The higher the SES score, the higher the objective SES of the subject [46].

### 2.3. Procedures

The study was reviewed by the school Ethics review committee, and the recruitment process ensured that each student and his/her parents who were interested in participating understood the intent of the study and completed the informed consent form. Trained psychology teachers at each school organized students to complete a series of questionnaires, including the Parental Psychological Control Scale, the Child Social Anxiety Scale, the Prosocial Behavior Scale, and the SES Scale. There was no special requirement for the order of filling in the questionnaire, and the whole test process took 15–20 min.

### 2.4. Data Analysis

SPSS 26.0 was used for descriptive statistical analysis of the data, and a freely available macro for SPSS (named Process Version 3.4, SPSS Inc., Chicago, IL, USA) was used to test the moderated mediation model.

## 3. Results

### 3.1. Common Method Bias

In this study, all data were self-reported by the subjects. In order to avoid the possibility that the estimation of one variable on other variables may be affected by Common Method Variance (CMV), Harman univariate factor analysis was used to perform the common method bias test [47]. The results showed that the first factor could explain 28.15% of the variance variation, which was lower than the critical criterion of 40% and satisfied the condition of statistical analysis. Therefore, it can be considered that there was no serious common method bias problem in this study.

### 3.2. Descriptive Statistical Analysis

The results of the descriptive statistical analysis are shown in Table 1. Parental psychological control was negatively correlated with children’s social anxiety, prosocial behavior, and SES, and social anxiety was negatively correlated with children’s prosocial behavior. In addition, there was no significant correlation between SES and children’s prosocial behavior, so SES was suitable to be used as a moderating variable for the subsequent moderating effect test.

### 3.3. The Mediating and Moderating Effects

Data were analyzed for mediating and moderating effects using SPSS Process (Version 3.4). Under the control of children’s gender and age, the mediating effect of social anxiety on the relationship between parental psychological control and children’s prosocial behavior was firstly analyzed. The 95% confidence intervals for mediating and moderating effects were estimated by 5000 samples to test the constructed moderating model. If the confidence interval did not include 0, the effect was significant; otherwise, the effect was insignificant. Social anxiety played a mediating role in the relationship between parental psychological control and children’s prosocial behavior. The indirect effect value was −0.0078, and the 95% confidence interval of the effect value was [−0.0012, −0.004], *p* < 0.05.

Then, the moderating effect of SES on the relationship between parental psychological control and prosociality was tested. The results showed that parental psychological control positively predicted social anxiety (*β* = 0.12, *p* < 0.001), and the product of parental psychological control and SES had a significant effect on prosociality (*β* = 0.008, *p* < 0.05), indicating that SES played a moderating role between parental psychological control and prosociality. Social anxiety could significantly predict children’s prosocial behavior levels (*β* = −0.067, *p* < 0.001). The specific results are shown in Table 2.

When SES was averaged, the Bootstrap 95% confidence interval was [−0.030, −0.007], excluding 0, which indicated that living in families with average SES, parental psychological control significantly impacts children’s prosocial behavior. When Z was at a low level (Z = 4.280), the Bootstrap 95% confidence interval was [−0.043, −0.013], excluding 0, which indicated that when SES was at a low level, parental psychological control had a significant impact on children’s prosocial behavior level. When Z was at a high level (Z = 7.141), the Bootstrap 95% confidence interval was [−0.024, 0.005], including 0, which indicated that the influence of parental psychological control on children’s prosocial behavior level was not significant in families with high SES. The specific results are shown in Table 3.

In conclusion, the prosocial behaviors of children living in high SES families were found to be not significantly affected by parental psychological control, while the prosocial behaviors of children living in middle and low SES families were found to be significantly negatively affected by parental psychological control. Therefore, the moderating effect of SES on the relationship between parental psychological control and children’s prosocial behavior exists.

Further simple slope analysis found (Figure 2) that when the SES was low, parental psychological control could negatively predict children’s prosocial behavior (*β* = −0.28, *p* < 0.001). When the SES was higher, the relationship between parental psychological control and prosocial behavior was not significant (*β* = −0.0091, *p* > 0.5). This indicates that the negative effect of parental psychological control on children’s prosocial behavior levels decreases with increasing SES.

## 4. Discussion

This study examined the relationship between parental psychological control and children’s prosocial behavior and examined the mediating effect of social anxiety and the moderating effect of SES. The findings of this study are helpful to better understand the prosocial behavior of Chinese children from several aspects and are of great significance for improving the level of prosocial behavior of children. Firstly, parental psychological control was positively correlated with children’s social anxiety and negatively correlated with children’s prosocial behavior. Secondly, social anxiety plays a partial mediating role between parental psychological control and children’s prosocial behavior. Finally, SES was a moderating variable for the relationship between parental psychological control and children’s prosocial behavior.

### 4.1. The Relationship between Parental Psychological Control and Children’s Social Anxiety and Prosocial Behavior

In this study, the influence of parental psychological control on children’s positive developmental outcomes (prosocial behavior) and negative developmental outcomes (social anxiety) was investigated simultaneously. Consistent with previous results, this study found a significant positive correlation between parental psychological control and children’s social anxiety [48,49] and children’s prosocial behavior [10]; that is, the higher the score of parental psychological control, the higher the score of children’s social anxiety and the lower the score of prosocial behavior. The above results were expected to be consistent, so Hypothesis 1 and Hypothesis 2 of this study were proved.

According to self-determination theory, human behavior is governed by different motivations, which are distributed along a continuum from autonomous to controlled motivations [15]. In the case of autonomous motivation, individual behavior is completely out of their own needs and intrinsic motivation and can achieve self-value to a great extent and facilitate happiness to the greatest extent. Under the condition of controlled motivation, individuals are forced to perform certain behaviors under pressure or obligation. Such internal and external pressure will promote individuals to have psychosocial maladjustment [50]. Specifically, compared with autonomy support, the parenting style of parental psychological control is more likely to lead to frustration in children’s autonomy motivation [50]. Under the high intensity of parental psychological control (such as guilt induction and withdrawal of love), on the one hand, children will have a kind of dependence and inertia in emotional control and thinking, which limits their independent space for growth and inhibits their attempts to develop in their own direction [51].

On the other hand, it may cause children to lose their independence and then lead to children’s social development disorders such as internalization symptoms, abnormal peer relations, and relationship aggression [52,53]. The results of a meta-analysis found a significant negative association between authoritarian parenting and prosocial behavior in children, which was stable across age groups (infancy, childhood, and adolescence) and across cultures (individualistic and collectivist cultural backgrounds) [54].

In the Chinese social environment, children gradually accept and internalize the basic values of collectivist interdependence and family obligations (such as respect for parents and filial piety) during socialization, and parental psychological control is often considered a proper parenting style [55]. However, the negative consequences caused by this parenting style have been largely ignored by researchers. This study is of great significance to the application and expansion of self-determination theory. On the one hand, it provides new evidence from the cultural background of Chinese collectivism for the research findings in this field, and on the other hand, it also confirms the cross-cultural adaptability of the theory.

### 4.2. Parental Psychological Control and Prosocial Behavior: Mediating Effects of Social Anxiety

Firstly, this study found that parental psychological control can significantly positively predict children’s social anxiety levels (Hypothesis 3), which is consistent with previous findings [24]. Past research found that children living in households where parents adopt an overly controlling and less autonomously supportive parenting style have higher levels of self-reported social anxiety symptoms [54]. Parents with high psychological control usually provide too much guidance and intervention to their children, which seriously limits their child’s opportunities for independent and free activities. Under the guidance of such a negative parenting style, children cannot receive proper support in the process of exploring new social relationships, and they are full of an unknowable and uncontrollable fear of social activities, which leads to the emergence of social anxiety symptoms [24].

Secondly, social anxiety can be negatively associated with children’s prosocial behavior, which is consistent with previous research results [28]. Related studies in the field of psychology have found that individuals with social anxiety generally have more difficulty in establishing meaningful interpersonal relationships than those without social anxiety [56]. In the process of social interaction, individuals with a high level of social anxiety rarely take the initiative to initiate interactive activities or participate in conversations, which leads to damage to their interpersonal relationships [57].

Thirdly, this study also found that social anxiety plays a mediating role between parental psychological control and children’s prosocial behavior; that is, parental psychological control will increase children’s social anxiety and then reduce children’s prosocial behavior level, which is consistent with previous research findings [18]. Barber et al. (2005) [19] conducted a systematic analysis of relevant theories and empirical studies in the field of parent–child relationships and found that various behaviors of parents and children can be incorporated into the framework of “parenting”. The core content of the framework is composed of two parts: supportive and controlling parenting styles [19]. Controlling parenting causes children to focus more on themselves and less on others, and to develop emotional and behavioral disorders during social interaction, thus, reducing prosociality [19].

The result of the study can be explained by affective event theory, which reckons the affective events (e.g., parental psychological control) experienced by individuals are related to their affective reaction (e.g., social anxiety), resulting in an individuals’ behavior (prosocial behavior). Family can be regarded as the initial environment for children’s growth, and strong parental psychological control will cause children to produce a series of negative emotions such as anxiety and depression. For example, in a 6-month longitudinal study involving Chinese and American families, Wang et al. (2007) [39] found that parental psychological control predicted suppressed emotional functioning in children. In addition, the research of Li et al. (2021) [58] also verified this view; namely, parental psychological control has a significant impact on children’s social anxiety, which includes the direct effect of parental psychological control on children’s social anxiety as well as the indirect effect through self-esteem and intolerance of uncertainty. As a kind of negative emotional experience, social anxiety impairs children’s social ability and makes children feel shy, nervous, and even afraid when contacting others. They often have a negative view of themselves and think others see them the same way. Therefore, they usually choose to avoid social interactions and suppress their true opinions and emotions to avoid interpersonal conflicts [59]. This kind of social avoidance limits the development of children’s social understanding ability and makes them experience social difficulties. In addition, socially anxious adolescents are prone to misunderstand the hostile intentions of others, negatively evaluate the behaviors of others [43], and have difficulty perceiving prosocial behaviors from others, which eventually leads to a vicious cycle of less prosocial behaviors and less prosocial behaviors received.

### 4.3. Parental Psychological Control and Prosocial Behavior: Moderating Effects of SES

SES includes family income, parental education, occupational status, etc. [35], which are important protective resources in children’s development. This study found that SES moderated the effect between parental psychological control and prosocial behavior. Specifically, at low SES, the stronger the degree of psychological control of parents, the lower the level of prosocial behavior of children; however, when the SES is high, there is no significant relationship between parental psychological control and children’s prosocial behavior level, which is the proof of Hypothesis 4.

One reason may be that parents of different socioeconomic statuses have different parenting styles, which affect children’s prosocial behavior levels. It is shown that there are significant class differences in parenting styles in families with various SESs. Parents with higher SES tend to adopt authoritative and permissive parenting styles, while parents with lower SES tend to choose authoritarian or neglectful parenting styles [37]. Therefore, parents with higher economic status not only require their children to have high standards but also guide their children in reasonable ways through the way of strict kindness so that they can calmly face the achievements and setbacks in their growth, have better development in mental health, and high levels of prosocial behavior. For parents with low economic status, based on the confinement of educational concepts such as “no beating, no success”, respect the traditional “stick” education, and use scolding or even beating instead of effective communication, so that children form blind obedience or suppression of indifference personality, fear to communicate with others, and low levels of prosocial behavior.

Another explanation of the moderating role of SES is that parents of different SES can provide different social resources for their children and, thus, affect their children’s prosocial behaviors. In families with higher SES, parents are able and conscious to cultivate and develop children’s non-cognitive abilities, such as taking children to participate in various parent–child activities, interest classes, and other social activities. Children who grow up in this environment are more confident and participate in social activities autonomously. Prosocial influence can increase the emotional return of giving [4]. Lyubomirsky et al. (2005) [3] also mentioned that using time and money to benefit others can allow the giver to obtain emotional rewards, and these children can obtain personal value and happiness beyond money by benefiting others [5].

## 5. Limitations and Further Direction

There are still some limitations in this study. First of all, this study was cross-sectional and, therefore, it was unable to determine the causal relationship between variables. Future studies may attempt longitudinal follow-up studies to explore more definitive causal relationships between variables. Secondly, this study did not distinguish between the psychological control of fathers and mothers. Some studies have shown that fathers play a more significant role in the social adjustment of children than mothers. Future studies can explore how the psychological control of fathers and mothers, respectively, affect the prosocial behavior of children. Thirdly, the subjects of this study were limited to primary school students in Beijing. Due to regional particularity, the generalizability of the results needs to be verified in more regions. In addition, this study paid attention to examining the association between parental psychological control and children’s prosocial behavior. Existing studies found that attachment in parent–child relationship can predict children’s prosocial behavior as well [60,61]. Thus, it is worthwhile to examine other predictors (e.g., attachment) of children’s prosocial behaviors in further studies.

## 6. Conclusions

Echoing calls by other researchers [62], parenting style plays an essential role in the internalization of prosociality in children. In the present study, our findings shed new light on both the specificity and commonality of the links between parental psychological control, social anxiety, SES, and prosocial behavior in childhood. This study has shown that the level of parental psychological control is shown to negatively predict children’s prosocial behaviors. The higher the degree of psychological control, the less prosocial behaviors the children show. Social anxiety partially mediates the relationship between parental psychological control and children’s prosocial behavior. Parental psychological control not only directly affects children’s prosocial behavior but also reduces children’s prosocial behavior by exacerbating children’s social anxiety. The mediating effect of social anxiety on the influence of parental psychological control on children’s prosocial behavior is moderated by family SES. Compared with high SES families, the indirect effect has a greater impact on children from middle or low SES families.

## Figures and Tables

**Figure 1 ijerph-19-11691-f001:**
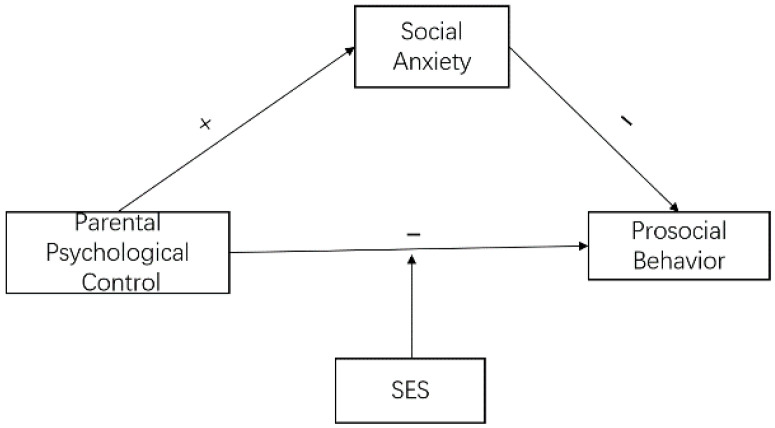
Research hypothesis model.

**Figure 2 ijerph-19-11691-f002:**
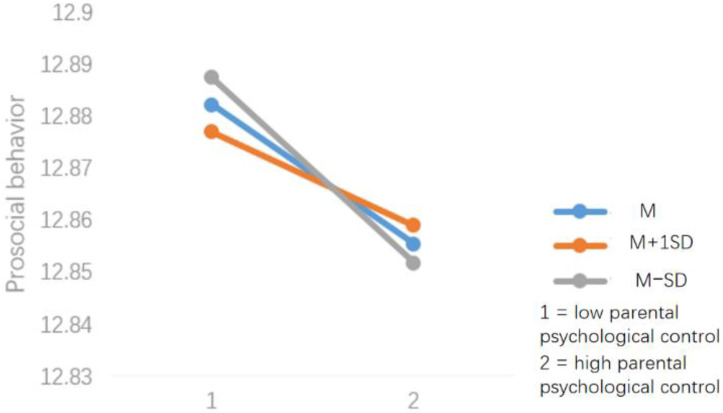
Simple slope test.

**Table 1 ijerph-19-11691-t001:** Results of descriptive statistical analysis.

Variables	M	SD	1	2	3	4
1. Parental psychological control	40.17	11.45	1			
2. Social anxiety	16.00	4.34	0.32 **	1		
3. Prosocial behavior	12.88	2.12	−0.15 **	−0.17 **	1	
4. SES	5.71	1.43	−0.08 *	0.00	0.01 *	1

Note. * *p* < 0.05, ** *p* < 0.01.

**Table 2 ijerph-19-11691-t002:** Test of moderated mediating effect.

	Prosocial Behavior				Social Anxiety			
	β	SE	t	*p*	β	SE	t	*p*
constant	16.129	0.907	17.781	0.000 **	11.172	0.485	23.043	0.000 **
parental psychological control	−0.056	0.021	−2.683	0.007 **	0.120	0.012	10.087	0.000 **
SES	−0.252	0.150	−1.675	0.094				
psychological control × SES	0.008	0.003	1.891	0.059				
Social anxiety	−0.067	0.015	−4.057	0.000 **				
*R^2^*	0.040				0.101			
Adjusted *R^2^*	0.036				0.099			
*F*	*F*(4,1197) = 12.491, *p =* 0.000				*F* (1,1200) =134.486, *p* = 0.000			

Note: ** *p* < 0.01.

**Table 3 ijerph-19-11691-t003:** Direct effects of different household SES.

Level	Level Score	Effect	SE	t	*p*	LLCI	ULCI
M − SD	4.280	−0.028	0.008	−3.618	0.000	−0.043	−0.013
M	5.710	−0.019	0.006	−3.256	0.001	−0.030	−0.007
M + 1SD	7.141	−0.009	0.007	−1.231	0.219	−0.024	0.005

Note: The level M − SD, M, M + SD represented high, average, and high SES separately. The level score was for the analytic sample of Chinese children. LLCI refers to the lower limit of the 95% range of the estimated value, and ULCI refers to the upper limit of the 95% range of the estimated value.

## Data Availability

The datasets generated and analyzed during the current study are available from the corresponding author on reasonable request.

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
