# Peer review of "Parental Psychological Control and Children’s Prosocial Behavior: The Mediating Role of Social Anxiety and the Moderating Role of Socioeconomic Status"

_ijerph, 2022, doi:10.3390/ijerph191811691_

Round 1

Reviewer 1 Report

The main goal of this study is to analyze the association between parental psychological control, social anxiety and socioeconomic status with 1202 Chinese children´s prosocial behavior. The manuscript fills an important gap in the literature by examining an understudied population of Chinese children and by analyzing some non-widely examined predictors of prosocial behavior, such as parental psychological control.

However, there are many aspects that need to be addressed and improved in the manuscript.

First, there needs to be put more emphasis on the current state of art and the main objective of the study, throughout the whole manuscript, and especially in the “Current study” section. Second, in both the abstract and the discussion, the self determination theory and problem-behavior theory are mentioned as being core theories in this investigation. Nevertheless, these are not addressed in the introduction, and it is not clear how these theories are connected to the conducted study. In addition, the affect event theory is explained in the discussion (line 339). However, this is not addressed in the introduction.

In both, the introduction and the discussion the association between the 5-HTTLPR genotype and prosocial behavior is mentioned. However, this biological link is not well explained nor introduced in the manuscript.

Also, given that parental psychological control is analyzed as a predictor of children´s prosocial behavior, the classical four parenting styles should be mentioned, as a result of the levels of parental affection and control. For example, the democratic parenting style in line 52 and 86 or the authoritarian and neglectful styles in lines 88-89. Also, Hoffman should be cited between lines 286-287, as he was the one that proposed the different parental disciplinary techniques (i.e., induction, love withdrawal and power assertion). It would also be interesting to include further information about the predictors of prosocial behavior (e.g., attachment), especially the familial ones, in order to introduce better the topic.

Moreover, socioeconomic status is analyzed as a moderator in the association between parental psychological control and prosocial behavior. However, this link and its moderating effect is not theoretically appropriately explained and addressed in the introduction nor in the discussion.  

Additionally, all the hypotheses should be theoretically supported both in the introduction and the discussion, and they should also be well formulated, in terms of what is/will be expected. Also, the type of expected associations (positive or negative) should be included in Figure 1.

Following the journal´s rules, all citations should be numbered, in the same way as in the references´list.

Finally, several edits are needed in the written English language and style, such as in verb tenses, and several sentences that seem confusing (e.g., lines 105-109, 240-245, the “Measurements” section, etc.).

Author Response

The main goal of this study is to analyze the association between parental psychological control, social anxiety and socioeconomic status with 1202 Chinese children´s prosocial behavior. The manuscript fills an important gap in the literature by examining an understudied population of Chinese children and by analyzing some non-widely examined predictors of prosocial behavior, such as parental psychological control.

However, there are many aspects that need to be addressed and improved in the manuscript.

First, there needs to be put more emphasis on the current state of art and the main objective of the study, throughout the whole manuscript, and especially in the “Current study” section. Second, in both the abstract and the discussion, the self determination theory and problem-behavior theory are mentioned as being core theories in this investigation. Nevertheless, these are not addressed in the introduction, and it is not clear how these theories are connected to the conducted study. In addition, the affect event theory is explained in the discussion (line 339). However, this is not addressed in the introduction.

Response: Thanks. We addressed the self-determination theory and affect event theory in the introduction and discussion. We put more emphasis on the current state of art and the main objective of the study.

In both, the introduction and the discussion the association between the 5-HTTLPR genotype and prosocial behavior is mentioned. However, this biological link is not well explained nor introduced in the manuscript.

Response: Thanks. We removed the association between the 5-HTTLPR genotype and prosocial behavior.

Also, given that parental psychological control is analyzed as a predictor of children´s prosocial behavior, the classical four parenting styles should be mentioned, as a result of the levels of parental affection and control. For example, the democratic parenting style in line 52 and 86 or the authoritarian and neglectful styles in lines 88-89. Also, Hoffman should be cited between lines 286-287, as he was the one that proposed the different parental disciplinary techniques (i.e., induction, love withdrawal and power assertion). It would also be interesting to include further information about the predictors of prosocial behavior (e.g., attachment), especially the familial ones, in order to introduce better the topic.

Response: Thanks. We added the classical four parenting styles and the relevant literatures. The predictors of prosocial behavior (e.g., attachment) was added in the further direction.

Moreover, socioeconomic status is analyzed as a moderator in the association between parental psychological control and prosocial behavior. However, this link and its moderating effect is not theoretically appropriately explained and addressed in the introduction nor in the discussion.  

Response: Thanks. The theoretical basis of moderating effect of SES was rewrote and discussed.

Additionally, all the hypotheses should be theoretically supported both in the introduction and the discussion, and they should also be well formulated, in terms of what is/will be expected. Also, the type of expected associations (positive or negative) should be included in Figure 1.

Response: Thanks. The theoretical basis of hypotheses was added. The associations (positive or negative) was marked in Figure 1.

Following the journal´s rules, all citations should be numbered, in the same way as in the references´list.

Response: Thanks. All citations were numbered.

Finally, several edits are needed in the written English language and style, such as in verb tenses, and several sentences that seem confusing (e.g., lines 105-109, 240-245, the “Measurements” section, etc.).

Response: Thanks. We invited a native speaker with PHD in the major of education to polish the language.

Reviewer 2 Report

Thank you for submitting this interesting paper, which is well written and presented.

There are some minor corrections of English, which I suggest in the following lines.

Line 31 should read an empirical study

Line 116/117 –First, most of the studies were conducted

Line 118-  It is therefore worthwhile….

Line 121 … in the existing studies.

Line 121/122 To address these gaps in research

Line 145 - …Chinese students.

Line 287 lower case ‘on’

Line 288 – emotional thinking

Line 339 – The effective… this sentence does not make sense.

Line 358 – perceiving

Line 369 … which supports Hypothesis 4.

Author Response

Thank you for submitting this interesting paper, which is well written and presented.

There are some minor corrections of English, which I suggest in the following lines.

Line 31 should read an empirical study

Line 116/117 –First, most of the studies were conducted

Line 118-  It is therefore worthwhile….

Line 121 … in the existing studies.

Line 121/122 To address these gaps in research

Line 145 - …Chinese students.

Line 287 lower case ‘on’

Line 288 – emotional thinking

Line 339 – The effective… this sentence does not make sense.

Line 358 – perceiving

Line 369 … which supports Hypothesis 4.

Response: Thanks. We revised them one by one.

Reviewer 3 Report

Greetings Author(s),

Overall, this is a worthwhile manuscript.  With a focus on prosocial behavior and most studies being conducted in western countries, this study examines the interplay of parental psychological control, prosocial behavior, and social anxiety.  The study also examines the role of SES in the relationship between parental psychological control and children’s prosocial behavior.  The research design and methods are appropriate, however clarifying the measures of internal consistency for employed scales will be beneficial.  Additionally, more discussion regarding Bronfenbrenner’s ecosystem theory or self-determination theory in relationship to the construct variables would benefit the introduction/methods.  Tables/Figures are supportive of the findings related in the manuscript, but additional clarifying labels will support reader clarification.  The conclusions drawn from the study are supported by the data as detailed in the paper.  Thank you for your efforts.

Additional considerations:

The reviewer recommends numbering references consistent with in-text citations.

Abstract: Reviewer recommends clarifying between parental psychological control or parental psychology control

Reviewer recommends using acronym PPC after 1st use

Line 53 Reviewer suggestion: (CEPS)2014 (space here)

Line 57 Reviewer suggestion: children’s (child’s) or children’s

Line 65 Reviewer suggestion: …found (remove that) moderate associations….

Line 101-09 Reviewer recommends clarification, perhaps breaking up the long sentence into several sentences

Line 121 Reviewer suggestion: Thirdly, the influence of SES is neglected in the existing study. (existing studies)

Reviewer suggestion: Label Figure 1 together with the picture and social anxiety runs over on the next line

Methods

Reviewer Question:  What were the published internal consistency of scales from previous literature

Line 153 Reviewer suggestion: a (space) 10-item

Line 166 Reviewer suggestion: missing information for the (coefficient of 0.82

Reviewer Question: Table 1 has a asterisk indicating significant association between SES and Prosocial behavior.  Is this significant?

Reviewer question: What is process 3.4?

Reviewer suggestion: Recommend clarifying Table 3.

Reviewer Question: What is 1 and 2 in Figure 2., slope test?

Line 287 Reviewer suggestion: ON the one hand, begins a new sentence

Line 313 Reviewer suggestion: childs’

Line 318 Reviewer suggestion: significantly negatively….

Line 358 Reviewer suggestion: perceiring should be perceiving?

Author Response

Overall, this is a worthwhile manuscript.  With a focus on prosocial behavior and most studies being conducted in western countries, this study examines the interplay of parental psychological control, prosocial behavior, and social anxiety.  The study also examines the role of SES in the relationship between parental psychological control and children’s prosocial behavior.  The research design and methods are appropriate, however clarifying the measures of internal consistency for employed scales will be beneficial.  Additionally, more discussion regarding Bronfenbrenner’s ecosystem theory or self-determination theory in relationship to the construct variables would benefit the introduction/methods.  Tables/Figures are supportive of the findings related in the manuscript, but additional clarifying labels will support reader clarification.  The conclusions drawn from the study are supported by the data as detailed in the paper.  Thank you for your efforts.

Additional considerations:

The reviewer recommends numbering references consistent with in-text citations.

Response: Thanks. All citations were numbered.

Abstract: Reviewer recommends clarifying between parental psychological control or parental psychology control

Response: Thanks. We revised it.

Reviewer recommends using acronym PPC after 1st use

Response: Thanks. We revised it.

Line 53 Reviewer suggestion: (CEPS)2014 (space here)

Line 57 Reviewer suggestion: children’s (child’s) or children’s

Line 65 Reviewer suggestion: …found (remove that) moderate associations….

Line 101-09 Reviewer recommends clarification, perhaps breaking up the long sentence into several sentences

Line 121 Reviewer suggestion: Thirdly, the influence of SES is neglected in the existing study. (existing studies)

Response: Thanks. We revised them one by one.

Reviewer suggestion: Label Figure 1 together with the picture and social anxiety runs over on the next line

Response: Thanks. We revised it.

Methods

Reviewer Question:  What were the published internal consistency of scales from previous literature

Response: Thanks. We added it.

Line 153 Reviewer suggestion: a (space) 10-item

Line 166 Reviewer suggestion: missing information for the (coefficient of 0.82

Response: Thanks. We revised it.

Reviewer Question: Table 1 has a asterisk indicating significant association between SES and Prosocial behavior.  Is this significant?

Response: Thanks. We revised it.

Reviewer question: What is process 3.4?

Response: Thanks. We revised it.

Reviewer suggestion: Recommend clarifying Table 3.

Response: Thanks. We clarified it.

Reviewer Question: What is 1 and 2 in Figure 2., slope test?

Response: Thanks. We clarified it.

Line 287 Reviewer suggestion: ON the one hand, begins a new sentence

Line 313 Reviewer suggestion: childs’

Line 318 Reviewer suggestion: significantly negatively….

Line 358 Reviewer suggestion: perceiring should be perceiving?

Response: Thanks. We revised them one by one.

Round 2

Reviewer 1 Report

The manuscript has been much more improved from teh last version. However, there are still some issues that need to be addressed. First, there are still several citations which are not numbered, such is in lines 60,64, 71, 73, 81, 310, 329, 342, 344, and 387. In addition, there arre several sentences which are incorrectly structured, and need to be clarified and better explained. For example, what doy meand when you mention "..controllin parenting would be associated with more need frustration" in lines 54-55? Line 92 should be "parental psychological control" and not parenting psychological controls. In line 117 should "Based on.." and not "Basing on.." In line 140 should be "Chinese students" in plural. The sentence in line 166 need to bet better explained, formulated and structured, as it is not clearly understood. The sentence between 401 and 403 should be rephrased so that it is understandable.

Author Response

The manuscript has been much more improved from the last version. However, there are still some issues that need to be addressed. First, there are still several citations which are not numbered, such is in lines 60, 64, 71, 73, 81, 310, 329, 342, 344, and 387.

Response: Thanks. All citations were numbered.

In addition, there are several sentences which are incorrectly structured, and need to be clarified and better explained. For example, what doy meand when you mention "..controllin parenting would be associated with more need frustration" in lines 54-55? Line 92 should be "parental psychological control" and not parenting psychological controls. In line 117 should "Based on.." and not "Basing on.." In line 140 should be "Chinese students" in plural. The sentence in line 166 need to bet better explained, formulated and structured, as it is not clearly understood. The sentence between 401 and 403 should be rephrased so that it is understandable.

Response: Thanks. We revised them one by one.
